# Influence of Mechanical Alignment on Functional Knee Phenotypes and Clinical Outcomes in Primary TKA: A 1-Year Prospective Analysis

**DOI:** 10.3390/jpm13050778

**Published:** 2023-04-30

**Authors:** Dominik Rak, Lukas Klann, Tizian Heinz, Philip Anderson, Ioannis Stratos, Alexander J. Nedopil, Maximilian Rudert

**Affiliations:** Orthopädische Klinik König-Ludwig-Haus, Lehrstuhl für Orthopädie der Universität Würzburg, 97074 Würzburg, Germany

**Keywords:** knee arthroplasty, mechanical alignment, clinical outcome, phenotype, level of evidence III, prospective study

## Abstract

In total knee arthroplasty (TKA), functional knee phenotypes are of interest regarding surgical alignment strategies. Functional knee phenotypes were introduced in 2019 and consist of limb, femoral, and tibial phenotypes. The hypothesis of this study was that mechanically aligned (MA) TKA changes preoperative functional phenotypes, which decreases the 1-year Forgotten Joint (FJS) and Oxford Knee Score (OKS) and increases the 1-year WOMAC. All patients included in this study had end-stage osteoarthritis and were treated with a primary MA TKA, which was supervised by four academic knee arthroplasty specialists. To determine the limb, femoral, and tibial phenotype, a long-leg radiograph (LLR) was imaged preoperatively and two to three days after TKA. FJS, OKS, and WOMAC were obtained 1 year after TKA. Patients were categorized using the change in functional limb, femoral, and tibial phenotype measured on LLR, and the scores were compared between the different categories. A complete dataset of preoperative and postoperative scores and radiographic images could be obtained for 59 patients. 42% of these patients had a change of limb phenotype, 41% a change of femoral phenotype, and 24% a change of tibial phenotype of more than ±1 relative to the preoperative phenotype. Patients with more than ±1 change of limb phenotype had significantly lower median FJS (27 points) and OKS (31 points) and higher WOMAC scores (30 points) relative to the 59-, 41-, and 4-point scores of those with a 0 ± 1 change (*p* < 0.0001 to 0.0048). Patients with a more than ±1 change of femoral phenotype had significantly lower median FJS (28 points) and OKS (32 points) and higher WOMAC scores (24 points) relative to the 69-, 40-, and 8-point scores of those with a 0 ± 1 change (*p* < 0.0001). A change in tibial phenotype had no effect on the FJS, OKS, and WOMAC scores. Surgeons performing MA TKA could consider limiting coronal alignment corrections of the limb and femoral joint line to within one phenotype to reduce the risk of low patient-reported satisfaction and function at 1-year.

## 1. Introduction

Successful TKA has been shown to substantially improve mobility and quality of life in patients with advanced osteoarthritis of the knee. The fact that the procedure of total joint replacement is one of the most successful and effective operations with excellent survivorship of the implants, there are still a considerable percentage of patients with ongoing problems with the replaced joint. Studies reveal that about 15–20% of patients are not satisfied with the implanted knee prosthesis [1,2]. Currently, most of these patients are treated with the mechanical alignment (MA) technique, which is still considered the “gold standard” in knee arthroplasty [3]. In MA, the orientation of the joint line and the bone resections are usually not evaluated and measured, as the resection thickness does not influence surgical decision making. However, one reason for the discomfort of a considerable proportion of the patients might be the change in the native orientation of the joint line and, consequently, a change in ligament balancing. To better evaluate the joint line and to understand how far this might influence the outcome, a lot of research and analysis was done, and the concept of phenotypes was established. This classification confirms a wide variability between individuals and challenges the standard of MA [4,5,6].

Functional phenotypes of the limb, femur, and tibia have been introduced by Hirschmann et al. to provide a contemporary classification of normal coronal limb and knee alignment [7,8]. Functional phenotypes categorize the hip-knee-ankle (HKA) angle, femoral mechanical angle (FMA), and tibial mechanical angle (TMA) within intervals of 3°. One remarkable finding of their work was that only 5.6% of non-osteoarthritic males and 3.6% of non-osteoarthritic females exhibit limb and knee alignment characteristics targeted by mechanically aligned (MA) total knee arthroplasty (TKA) [8].

MA sets the femoral component perpendicular to the femoral mechanical axis, which changes the pre-arthritic distal femoral joint line of most patients because the orientation of this line varies from −6° varus to 6° valgus relative to the femoral mechanical axis in the non-osteoarthritic knee [7,9,10,11,12]. When performing MA TKA in 84% of patients that have a pre-arthritic neutral or valgus femoral phenotype (FMA > 91.5°), the femoral phenotype will change by at least one category [7]. In addition, MA sets the tibial component perpendicular to the tibial mechanical axis, which changes the pre-arthritic tibial joint line in most patients because the orientation of this line varies from −9° varus to 3° valgus relative to the tibial mechanical axis in the non-osteoarthritic knee [7,13]. When performing MA TKA in 71% of patients with a pre-arthritic tibial joint orientation different from perpendicular to the tibial mechanical axis, the tibial phenotype will change at least one category [7].

The limb, femoral, and tibial phenotypes in osteoarthritic knees are differently distributed than in healthy non-arthritic knees, with higher deviations from neutral phenotypes [14]. Consequently, it can be expected that any alignment strategy in TKA will change most patients’ osteoarthritic limb, femoral, and tibial phenotype regardless of whether the target of the alignment strategy is to restore the patient’s pre-arthritic alignment, i.e., kinematic alignment (KA), or to create perpendicular joint lines in relation to the corresponding mechanical axis, i.e., MA [15].

Recently, a study concluded that the MA-induced change of the patient’s joint line obliquity and arithmetic HKA angle did not influence patient outcome [16]. This study used an alignment classification system termed coronal alignment of the knee (CPAK), which groups joint line obliquity and the arithmetic HKA angle (an angle computed from the femoral and tibial joint line orientation in relation to the mechanical axis of the corresponding bone) in nine phenotypes [17]. Because joint line obliquity is categorized (e.g., apex distal, neutral, apex proximal) and the arithmetic HKA angle is categorized (e.g., varus, neutral, valgus), it is not sensitive to changes within one category. In contrast, the classification system of functional limb, femoral, and tibial phenotypes is quantitative and, therefore, could help identify the magnitude of phenotype category change that lowers clinical outcome scores.

Assessing whether a change of functional phenotype adversely affects clinical outcomes requires knowing the minimum clinically important difference (MCID) difference of patient-reported questionnaires. The MCID is the smallest change in score that patients perceive as meaningful, which would cause clinicians to consider modifications in their treatment approach. For example, representative MCID values are 13 points for the Forgotten Joint Score (FJS), 3–5 points for the Oxford Knee Score (OKS), and 10 points for the Western Ontario and McMaster Universities Osteoarthritis Index (WOMAC) score [18,19,20].

A literature review found no reports of MA TKA that assessed the change in functional phenotype and whether the change adversely affected clinical outcome scores. Accordingly, this prospective study determined the proportion of patients with a change of functional limb, femoral, and tibial phenotype after MA TKA and which change caused a low 1-year Forgotten Joint (FJS) and Oxford Knee (OKS) and high WOMAC score. A follow-up period of one year seemed reasonable because knee function reaches a plateau within the first postoperative year and remains stable in the following years [21,22,23,24].

## 2. Materials and Methods

An institutional review board approved this prospective study (IRB-189/19). The lead author (DR) enlisted four experienced academic knee arthroplasty specialists who each supervised a cemented primary MA TKA on 20 consecutive patients with end-stage osteoarthritis using conventional manual instrumentation and a posterior cruciate ligament retaining implant design (Triathlon Stryker, Kalamazoo, MI, USA). Excluded were patients with avascular necrosis, septic arthritis, prior intra-articular fracture, or a severe pre-operative knee deformity that required revision components to restore stability. Patients with no pre-operative or postoperative long-leg radiographs (LLR) were also excluded. In addition, surgeons recorded pre-operative values of body mass index (BMI), knee extension, knee flexion, Oxford Knee Score (OKS), and Western Ontario and McMaster Universities Arthritis Index (WOMAC) (100 worst, 0 best) on each patient.

The following describes the key points of the surgical technique. After exposing the knee, the surgeon classified the primary location of the osteoarthritic as medial (i.e., varus deformity), lateral (i.e., valgus deformity), or patellofemoral. The distal femoral resection guide, set at 6° of valgus relative to a rod inserted into the intramedullary canal, was seated flush with the most distal condyle of the femur, which determined the varus-valgus (V-V) joint line orientation of the femoral component. The saw slot of an extramedullary tibial resection guide set perpendicular to the mechanical axis of the tibia was positioned 8 mm distal to intact cartilage on the tibial articular surface, which determined the V-V joint line orientation of the tibial component. The surgeon, at their discretion, resurfaced the patella and released ligaments to balance the TKA.

On LLR obtained before and three days after MA TKA, the following angles were computed on the operated limb: (1) The HKA angle measured between the lines connecting the centers of the femoral head, the knee, and the talus, (2) the FMA measured between the femoral mechanical axis and a tangent to the distal femoral condyles, (3) the TMA measured between the tibial mechanical axis and a tangent to the proximal tibia joint surface or the tibial baseplate. Each angle was assigned to a phenotype category [7,8,14].

One year postoperatively, the surgeon sent the FJS, OKS, and WOMAC questionnaires to each patient. Those that filled out each questionnaire were included in this study.

### Statistical Analysis

The Shapiro-Wilk test determined the normality of the dependent variables. The mean ± standard deviation (SD) and median and interquartile range (IQR) described normal and non-normal dependent variables (JMP Pro, 16.2.0, www.jmp.com, accessed on 23 February 2023). Based on pre- and postoperative HKA angle, FMA, and TMA measurements, patients were assigned to a pre- and postoperative phenotype category, and the change in phenotype category was computed. For each phenotype, the Wilcoxon/Kruskal–Wallis test determined the significance of the difference in the one-year FJS, OKS, and WOMAC scores between each change in the phenotype category. Significance was set at *p* < 0.05.

## 3. Results

Of the 83 eligible patients, 11 patients did not receive a postoperative LLR. Thirteen patients did not return the one-year clinical outcome questionnaires, leaving 59 patients for final data analysis. Table 1 shows the years each surgeon practiced TKA and the preoperative patient characteristics and function scores for those patients treated by each surgeon. There were no significant differences in the proportion of females to males, mean age, mean BMI, the proportion of varus, valgus, and patellofemoral deformities, and preoperative OKS and WOMAC scores between surgeons. Table 2 shows the one-year median FJS, OKS, and WOMAC scores were not significantly different between the patients treated by each surgeon. Hence, this study combined the patients for analysis.

The pre- and postoperative functional phenotype distribution is illustrated in Figure 1. Remarkably, 42% of the HKA phenotype and 41% of the FMA phenotype, but only 25% of the TMA phenotype were changed to more than one category by MA TKA (Figure 1).

For the one-year FJS, patients with a HKA phenotype change of more than one category had a significantly lower median score of 27 points relative to the 59-point score of those with no or only one category change (*p* = 0.0002) (Figure 2). In addition, patients with a FMA phenotype change of more than one category had a significantly lower median score of 28 points relative to the 69-point score of those with no or only one category change (*p* < 0.0001). A change in TMA phenotype had no effect on the FJS.

For the one-year OKS, patients with a HKA phenotype change of more than one category had a significantly lower median score of 31 relative to the 41-point score of those with a no or only one category change (*p* < 0.0001) (Figure 3). In addition, patients with a FMA phenotype change of more than one category had a significantly lower median score of 32 points relative to the 40-point score of those with no or only one category change (*p* < 0.0001). A change in TMA phenotype had no effect on the OKS.

For the one-year WOMAC, patients with a HKA phenotype change of more than one category had a significantly higher median score of 30 points relative to the 10-point score of those with no or only one category change (*p* = 0.0002) (Figure 4). In addition, patients with a FMA phenotype change of more than one category had a significantly higher median score of 24 points relative to the 8-point score of those with no or only one category change (*p* < 0.0001). A change in TMA phenotype had no effect on the WOMAC score.

To better analyze whether other factors or variables influence the postoperative outcome after total knee arthroplasty, we added a simple regression analysis that showed that only age, BMI, Preoperative OKS, and preoperative FMA influenced the postoperative outcome, which is already well-known in literature [25,26,27]. In addition, we performed a Student *t*-test to analyze if these variables have an influence or difference in the two groups displaying the change in phenotype categories. As displayed in Table 3 it showed that none of the variables that influence postoperative outcomes in general have a significant difference in the two groups, whether in the HKA, FMA, or TMA phenotype change group.

## 4. Discussion

The most important findings of the present study were that changing the patient’s functional limb and femoral phenotype by more than one category significantly lowered clinical outcome scores, while changing the patient’s functional tibial phenotype has a negligible effect on clinical outcome scores.

To the authors’ knowledge, this is the first study evaluating the impact of changing the patient’s functional limb and knee phenotype on clinical outcomes after MA TKA. This study confirms that a too vigorous change in a patient’s limb and knee alignment reflects poorly on clinical outcome [15,28,29,30]. The changes of the phenotype and, in that sense, a change of the joint line and, thereby, a change of the soft tissue tension may have a distinct impact on the short and long-term outcome. The change in soft tissue balancing is one key point in the discussion of different alignment strategies for knee arthroplasty. A different pressure distribution, soft tissue change, and change of joint alignment might lead to ligament imbalance which might be one reason for dissatisfaction after TKA [31,32,33].

To address that concern, multiple alternative alignment techniques have evolved, including unrestricted KA, which aims to restore the patient’s pre-arthritic alignment without limitations, and functional alignment, which aims to restore the pre-arthritic alignment within defined boundaries, while minimizing changes to the joint line orientation and soft tissue releases [34,35].

While the present study compared pre- and postoperative functional phenotypes (i.e., alignment), it did not evaluate how MA TKA changed patients’ pre-arthritic functional phenotypes. Osteoarthritis changes functional phenotype distribution with a wider deviation from neutral [14]. The patient’s pre-operative limb, femoral, and tibial phenotype is, therefore, not an adequate target when planning a TKA. However, the results from the present study indicate that patient-reported outcome scores are sensitive to a change of limb and femoral phenotype beyond one category, which could help surgeons performing functional alignment to further personalize their alignment boundaries by avoiding phenotype changes of more than one category [29].

An unexpected result from the present study is that a change in the tibial phenotype did not alter clinical outcome scores. This finding indicates that postoperative patient function is more sensitive to the restoration of pre-arthritic limb alignment and femoral joint line orientation, which corresponds to the concept of unrestricted KA [36]. A principle of unrestricted KA is to restore the flexion–extension (FE) axis of the tibia, which is located within the femoral condyles [9,37].

The present study can help understand why using robotic assistance in MA TKA does not improve patient-reported function beyond the threshold of MCID [38,39,40]. While robotic assistance or patient-specific instrumentation (PSI) improves the accuracy of component positioning to the MA TKA target, it does not necessarily result in substantial advantages concerning patient functionality [41,42,43]. Executing MA TKA with high accuracy consistently changes functional phenotypes in more than one category and consequently exposes the patient to an increased risk of inferior clinical outcomes.

The present study has several limitations. First, only one implant system was used to treat the patients in the present study. To generalize the results from this study, surgeons performing MA TKA with other implant systems could measure functional phenotypes and assess whether a change in functional phenotype beyond one category causes a drop in patient-reported outcome scores one year after TKA. Finally, the frequency and extent of soft tissue releases were not included in the data analysis of this study. Because alternative alignment techniques strive to reduce the frequency and extent of soft tissue releases during TKA, it might be of interest whether the change in functional phenotypes is associated with the frequency and extent of soft tissue releases and whether the soft tissue release itself influences patient-reported outcomes. Future studies shall answer this question.

## 5. Conclusions

When performing MA TKA, surgeons should recognize the adverse consequences of changing the patient’s presurgical limb and femoral phenotype by two or more categories, as this significantly lowered patient-reported outcomes after one year. If surgeons use robotic instrumentation or navigation to align a TKA, it would be worth considering setting alignment targets that avoid a phenotype change of more than one category.

## Figures and Tables

**Figure 1 jpm-13-00778-f001:**
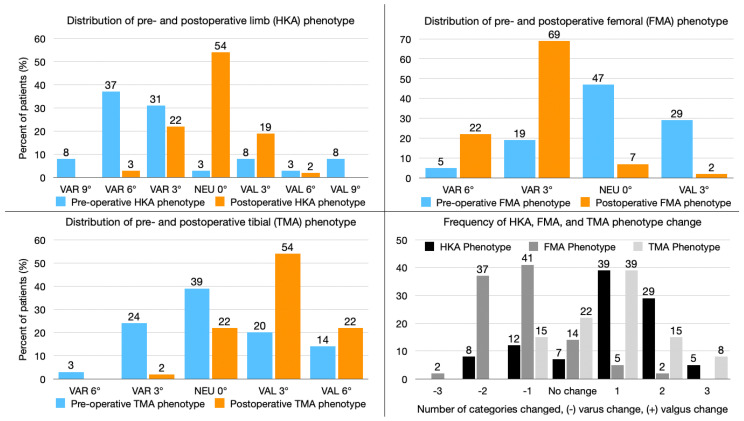
The composite shows the distribution of limb (**top left graph**), femoral (**top right graph**), and tibial (**bottom left graph**) phenotypes before and after mechanically aligned total knee arthroplasty (MA TKA). The number of categories MA TKA changed in each functional phenotype is depicted in the (**bottom right graph**).

**Figure 2 jpm-13-00778-f002:**
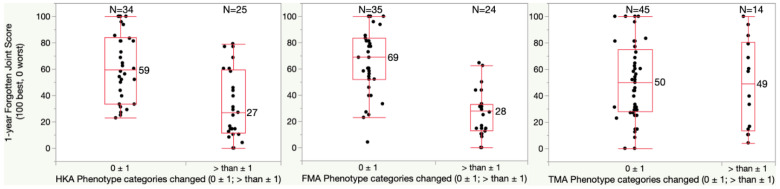
Boxplots show the Forgotten Joint Score (FJS) (100 best, 0 worst) of patients who had a phenotype change of fewer than two categories and of more than one category. The median FJS of patients with more than 1 category change of the limb (HKA) and femoral (FMA) phenotype was significantly lower by at least 2 times the 13-point MCID relative to patients whose phenotype changed only one category or less (*p* = 0.0002 (HKA) and <0.0001 (FMA)). A change of the tibial (TMA) phenotype of more than one category was less frequent and did not lower the FJS.

**Figure 3 jpm-13-00778-f003:**
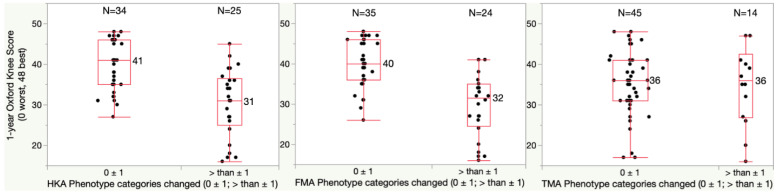
Boxplots show the Oxford Knee Score (OKS) (48 best, 0 worst) of patients who had a phenotype change of fewer than two categories and of more than one category. The median OKS of patients with more than 1 category change of the limb (HKA) and femoral (FMA) phenotype was significantly lower by at least once the 5-point MCID relative to patients whose phenotype changed only one category or less (*p* < 0.0001). A change of the tibial (TMA) phenotype of more than one category was less frequent and did not lower the OKS.

**Figure 4 jpm-13-00778-f004:**
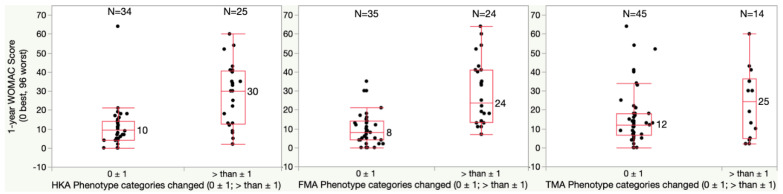
Boxplots show the WOMAC score (0 best, 96 worst) of patients who had a phenotype change of fewer than two categories and of more than one category. The median WOMAC score of patients with more than 1 category change of the limb (HKA) and femoral (FMA) phenotype was significantly higher by at least once the 10-point MCID relative to patients whose phenotype changed only one category or less (*p* = 0.0002 (HKA) and <0.0001 (FMA)). A change of the tibial (TMA) phenotype of more than one category was less frequent and did not increase the WOMAC score.

**Table 1 jpm-13-00778-t001:** Pre-operative patient characteristics and function scores for the patients treated by each surgeon.

	Surgeon 1	Surgeon 2	Surgeon 3	Surgeon 4	*p*-Value
Years of Practice	30	22	17	14	
	**Patients’ Preoperative Characteristics and Function Scores**				
	**Number of Patients (N)**				
Female/Male	6/10	8/7	8/7	7/6	*p* = 0.7520 **
Preoperative Deformity: Varus/Valgus/Patellofemoral	12/4/0	12/2/1	13/2/0	11/2/0	*p* = 0.8108 **
	**Mean ± Standard Deviation**				
Age (years)	67 ± 9	68 ± 7	63 ± 7	66 ± 13	*p* = 0.5266 *
BMI ^1^	29 ± 8	33 ± 6	31 ± 5	33 ± 8	*p* = 0.3121 *
Knee Extension (deg)	3 ± 4	4 ± 6	2 ± 3	3 ± 4	*p* = 0.6768 *
Knee Flexion (deg)	112 ± 10	111 ± 8	104 ± 11	105 ± 10	*p* = 0.0597 *
Oxford Knee Score (48 best, 0 worst)	21 ± 4	20 ± 5	19 ± 5	23 ± 6	*p* = 0.1875 *
WOMAC ^2^ (0 best, 96 worst)	43 ± 14	48 ± 12	49 ± 18	45 ± 19	*p* = 0.5935 *

^1^ Body-Mass-Index; ^2^ Western Ontario and McMaster Universities Arthritis Index; * ANOVA determined differences between surgeons; ** Pearson’s Chi-Square Test determined differences between surgeons.

**Table 2 jpm-13-00778-t002:** Postoperative patient function scores for the patients treated by each surgeon.

	Surgeon 1	Surgeon 2	Surgeon 3	Surgeon 4	*p*-Value
Number of patients (N) with 1-year follow-up	16	15	15	13	
	**Median and [IQR] ^1^ of Postoperative Function Scores**				
Forgotten Joint Score (100 best, 0 worst)	45 [31 to 73]	60 [15 to 69]	33 [25 to 77]	44 [28 to 89]	*p* = 0.4692 *
Oxford Knee Score (48 best, 0 worst)	35 [31 to 45]	37 [27 to 41]	40 [27 to 45]	36 [31 to 43]	*p* = 0.3257 *
WOMAC ^2^ (0 best, 96 worst)	13 [6 to 16]	17 [9 to 41]	11 [4 to 22]	13 [4 to 28]	*p* = 0.8263 *

^1^ Interquartile Range [IQR] ^2^ Western Ontario and McMaster Universities Arthritis Index; * Kruskal-Wallis Test determined differences between surgeons.

**Table 3 jpm-13-00778-t003:** Analysis of the distribution of independent variables that influence postoperative outcomes with respect to the two Phenotype change groups.

	HKA Phenotype Categories Change 0 ± 1	HKA Phenotype Categories Change More Than ± 1	*p*-Value
	Mean ± Standard Deviation		
Age	67.2 ± 8.9	64.7 ± 9.1	*p* = 0.2927 *
BMI	31.7 ± 7.4	31.6 ± 6.0	*p* = 0.9712 *
Preop Oxford	21.2 ± 4.2	19.8 ± 5.7	*p* = 0.3138 *
Preop FMA	92.7 ± 2.0	92.6 ± 2.6	*p* = 0.9387 *

* *t*-Test determined differences between the two groups who had a phenotype change of fewer than two categories and of more than one category.

## Data Availability

The data sets to support the findings of this study are included within the article, including figures and tables. Any other data used to support the findings of this study are available from the corresponding authors upon request.

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
