# Peer review of "Influence of Mechanical Alignment on Functional Knee Phenotypes and Clinical Outcomes in Primary TKA: A 1-Year Prospective Analysis"

_jpm, 2023, doi:10.3390/jpm13050778_

Round 1

Reviewer 1 Report

This study prospectively explored the effect of knee phenotypes change on knee function scores before and after TKA surgery in 59 patients and found that the functional limb and femoral phenotypes change of more than 1 type would result in low patient satisfaction. The study is well done and clinically instructive.

However, there are many factors that could affect patient satisfaction and knee function scores after TKA, including prosthesis position (coronal and sagittal), tourniquet time, degree of soft tissue release, and postoperative rehabilitation et al., in addition to lower extremity alignment, surgeon, and implant system. Did the authors take into account the influence of the above factors?

Line 14 The study finally included valid data from 59 patients, and it is suggested to mark only the number of valid samples in the abstract and also to avoid duplication with line 20.

Line 86 It is suggested that the authors add clinical study registration information.

Line 128-129 It is suggested to avoid the duplication of the "Results" content and the "Table" content.

Line 192 It is suggested that the discussion section should be more adequate and include a discussion of the reasons for changes in clinical outcomes due to phenotypic changes. Is it because of pain, the change in gait, or the change in pressure distribution? There are many aspects included in the scores, can a more detailed comparison and discussion be made.

Line 291 To my knowledge, many studies have demonstrated that the distal femoral joint orientation is not perfectly parallel to the flexion-extension axis. It may be controversial for the author to express it in this way.

The quality of English writing was not found to be any problem.

Author Response

Dear honorable reviewers and editors

Please find enclosed our manuscript entitled ”Influence of Mechanical Alignment on Functional Knee Phenotypes and Clinical Outcomes in Primary TKA: A 1-Year Prospective Analysis”, which we resubmit for consideration of publication in JPM.

We thank you the editors and reviewers for reading our manuscript and providing suggestions to clarify the study's message. Please find below our point-by-point response to the reviewers’ concerns.

Thank you for coordinating the review of our manuscript. Very truly yours,

Dominik A. Rak, M.D.
Department of Orthopaedic Surgery, König-Ludwig-Haus, University of Würzburg

Reviewer 2 Report

This is a well designed and analyzed study. There are few suggestions and comments which the authors can address

1) A detailed literature study on prior art can be added in the introduction section. 

2) Since it is a one year prospective study, the authors can add a section on what morphological changes they may anticipate in a long term study and explain the biomechanical causes for the same.

3) In future work the authors can analyze the morphology of femoral and tibial components such as cartilage height, bone angles, notch dimensions,  condyle distances etc

Author Response

(The authors gave the same response as above.)
